# Assessment of Moraine Cliff Spatio-Temporal Erosion on Wolin Island Using ALS Data Analysis

Marcin Winowski [1,*], Jacek Tylkowski [1] and Marcin Hojan [2]

[1] Institute of Geoecology and Geoinformation, Faculty of Geographical and Geological Sciences, Adam Mickiewicz University, Krygowski 10, 61-680 Poznan, Poland; jatyl@amu.edu.pl

[2] Department of Landscape History Research, Institute of Geography, Kazimierz Wielki University, Kościeleckich Square 8, 85-033 Bydgoszcz, Poland; homar@ukw.edu.pl

[*] Correspondence: marwin@amu.edu.pl

**Abstract:** The aim of the article is to present the temporal and spatial variability of the cliff coast erosion of the Wolin Island in 2012–2020 in three time periods (2012–2015, 2015–2018, 2018–2020). The research used data from airborne laser scanning (ALS), based on which DEM models were made. Based on the differences between the models, the amount of sediment that was eroded by the sea waves was determined. The conducted research showed that, in the analyzed period, the dynamics of the Wolin cliffs were characterized by high variability. The greatest erosion was observed on sandy cliffs, and the smallest on clay cliffs and on cliffs that are densely covered with vegetation. In the sediment budget studies, two seashore erosivity indicators were proposed: length-normalized sediment budget ($L_B$) ($m^3/m$) and area-normalized sediment budget ($A_B$) ($m^3/m^2$). The average annual dynamics of the cliff edge erosion on the Wolin Island was found to be $L_B = 6.6 \pm 0.3\ m^3/m/a$, $A_B = 0.17 \pm 0.01\ m^3/m^2/a$. The results obtained are comparable with other postglacial cliffs. The use of the differential analysis of DEM models allows for the determination of the dynamics of the cliff coast and may be used in spatial development and planning of seashore protection zones.

**Keywords:** differential analysis; moraine cliff coast; DEM models; Southern Baltic; cliff morphodynamics; erosivity indexes





## 1. Introduction

The increase in air temperature, which we observe in the world today, causes the accelerated melting of ice caps and the rise of the level of seas and oceans, including the Baltic Sea by 0.01–0.02 m [1,2]. In addition, in mid-latitudes, climate changes affect changes in the trajectory and frequency of movement of the lows from the W and NW sectors towards the E and SE sectors. On the Polish coast of the South Baltic, it leads to the formation of storm surges, causing increased strength of sea erosion in the coastal zone [1,3–15] especially those of an extreme nature [16].

The analysis of hydrometeorological data has shown that on the Polish coast of the South Baltic Sea there are long-term trends favoring the erosion of both the cliff and dune coasts [17]. Research is being carried out on volumetric changes in sea coasts caused by erosion and accumulation processes. Most often, these are studies relating to short time intervals and short fragments of the coast with lengths of 0.4 to 2.5 km [4,5,8,18–25]. Research is very rarely conducted on a long stretch of coast, e.g., 18 km [21,23,26]. Test results are often given in different units, which makes them incomparable. Therefore, the presented article proposes two coast erosivity indicators, which make it possible to conduct comparative studies for cliffs around the world.

Research on erosion and accumulation in the coastal zone of the seas is very important in the context of marine coastal zone management. This is especially true of areas threatened by erosion and the use of methods to protect the seashore in these areas [27–35]. The issue of the dynamics of cliffs is a frequently discussed problem in the literature [19,21–24,36–42].

In the second half of the 20th century, comparative analysis of archival material was most often used, and simple measurements were made to determine the rate of recession of the cliffs [1,4,8,43,44]. Estimating the rate of receding the cliffs is a very important issue of research in the context of managing the coastal zone, geohazards, and preventing natural disasters [42,45–51]. However, it should be borne in mind that the recession of the cliff is the result of many components that are impossible to capture in two-dimensional cartographic and photogrammetric analyses. In recent years, there has been a dynamic development of remote registration methods, which made it possible to monitor coasts using the airbone laser scanning (ALS), terrestrial laser scanning (TLS), and mobile laser scanning (MLS) methods, as well as structure from motion–unmanned aerial vehicle (SfM-UAV) photogrammetric methods [4,19,25,47,52–64]. Coastal monitoring by remote recording allows for an accurate calculation of the sediment budget and the detection of areas susceptible to erosion. The three-dimensional data from laser scanning should be considered the most valuable.

Therefore, the main objective of the following studies was to determine the temporal and spatial differentiation of the erosion of the cliff coast of the Wolin Island built of unconsolidated moraine sediments using the ALS data for the eight-year research period 2012–2020.

The temporal dynamics of the cliff erosion were determined in the context of sea hydrodynamics. Erosive storm surges which caused the greatest loss of cliff sediments were indicated. For the studied period, the main episodes of shore erosion related to exceeding the erosive threshold value for the maximum sea level were indicated, as well as the energy of storm surges, the wave direction, and the height of significant waves. In addition, the spatial differentiation of the cliff erosion was determined, which made it possible to identify sections of the cliff endangered by increased erosion and sections with relatively greater resistance to degradation.

## 2. Materials and Methods

### 2.1. Research Area

The coast of the Southern Baltic Sea on the Wolin Island is classified as a cliff (Figure 1) and a dune type. The cliffs of the Wolin Island were formed as a result of the erosion of the end moraine, which was formed during the retreat of the last Pleistocene glaciation. The end moraine is made up of glacial clays, fluvioglacial sands, and muds associated with local pools. On the surface of the moraine there are aeolian cover sands of varying thickness, and locally there are small dunes (Figure 2). The diverse geological structure of the Wolin cliffs causes different rates of cliff erosion. The cliff coast of the Wolin Island in the Pomeranian Bay zone consists of two sections: western and eastern. In these sections, there are sectors composed of sand, sectors composed of clay, and sectors with a mixed structure—sand and clay. In the case of sandy loam sectors, the lower part of the sector is made of clay and the upper part is made of sand. Both clay and sandy sediments are of variable thickness. The height of the cliffs reaches a maximum of 95 m a.s.l.

### 2.2. Data Sources

In order to determine the temporal and spatial differentiation of the erosion of the moraine cliffs of the Wolin Island, data on the topography of the studied section of the sea coast were collected and processed. The morphodynamics of the cliffs are primarily determined by the hydrodynamic sea conditions; therefore, the interpretation of spatial diversity and the dynamics of erosion of the moraine cliff of Wolin Island would not be possible without taking into account the sea level, wave height, and the direction of its approach to the coast.

The volume changes of the cliff were analyzed for the eight-year research period 2012–2020 in three time periods 2012–2015, 2015–2018, 2018–2020 (to capture the temporal variability of erosion processes) and as a whole (2012–2020). The research objective was achieved on the basis of the analysis of the altitude data characterizing the landform. Altitude data were obtained from point clouds obtained from airborne laser scanning (ALS).

Airborne laser scanning is most suitable for mapping long sections of the coast because it provides data with high spatial resolution and with a relatively low point position error.

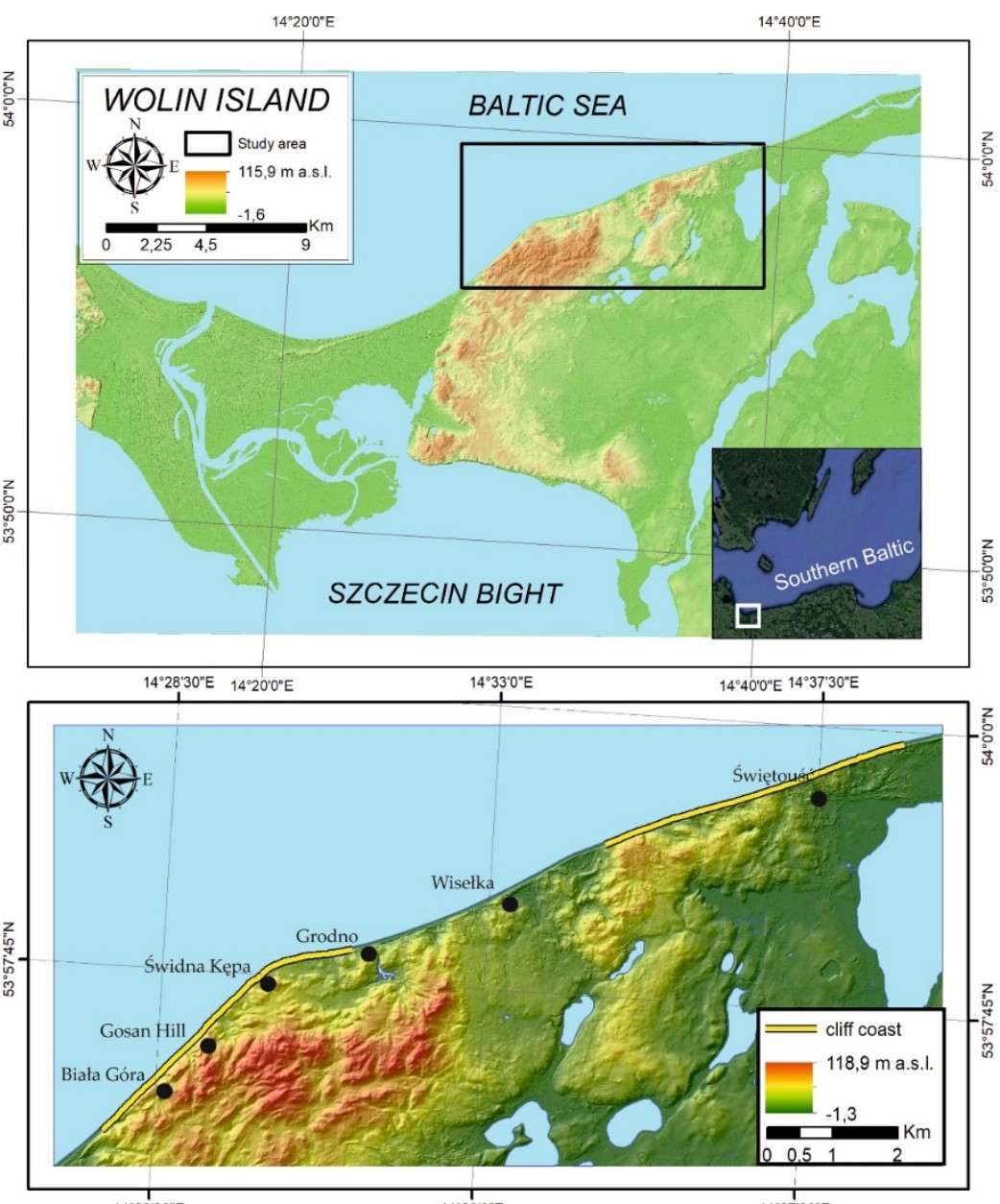

**Figure 1.** Study area. The map shows the western and eastern section that have been subjected to erosion analyses (Source: this study).

ALS data were obtained from the Maritime Office in Szczecin, which systematically commissions measurements as part of the monitoring of the Baltic coast in Poland. The scanning covered a 0.5 km wide coastal strip (from the water line inland). Each of the campaigns took place in the fall season (Table 1). The coastal relief data were used for a section of 13.9 km (start: N 53.940684, E 14.463470, end: N 53.998392, E 14.644526) (Table S1). The number of points of the analyzed clouds from ALS ranged from 216,253,380 to 492,283,200. The point cloud density ranged from 3 pts/m² to 8 pts/m² (Table 1). Each of the acquired clouds were classified by data provider according to the American Society for Photogrammetry and Remote Sensing (ASPRS) standard [65]. Unfortunately, the data provider did not provide information on the accuracy of the point clouds. Therefore,

the accuracy of volumetric analyses assessment was based on reference points analysis (explained in 2.3 GIS analysis section).

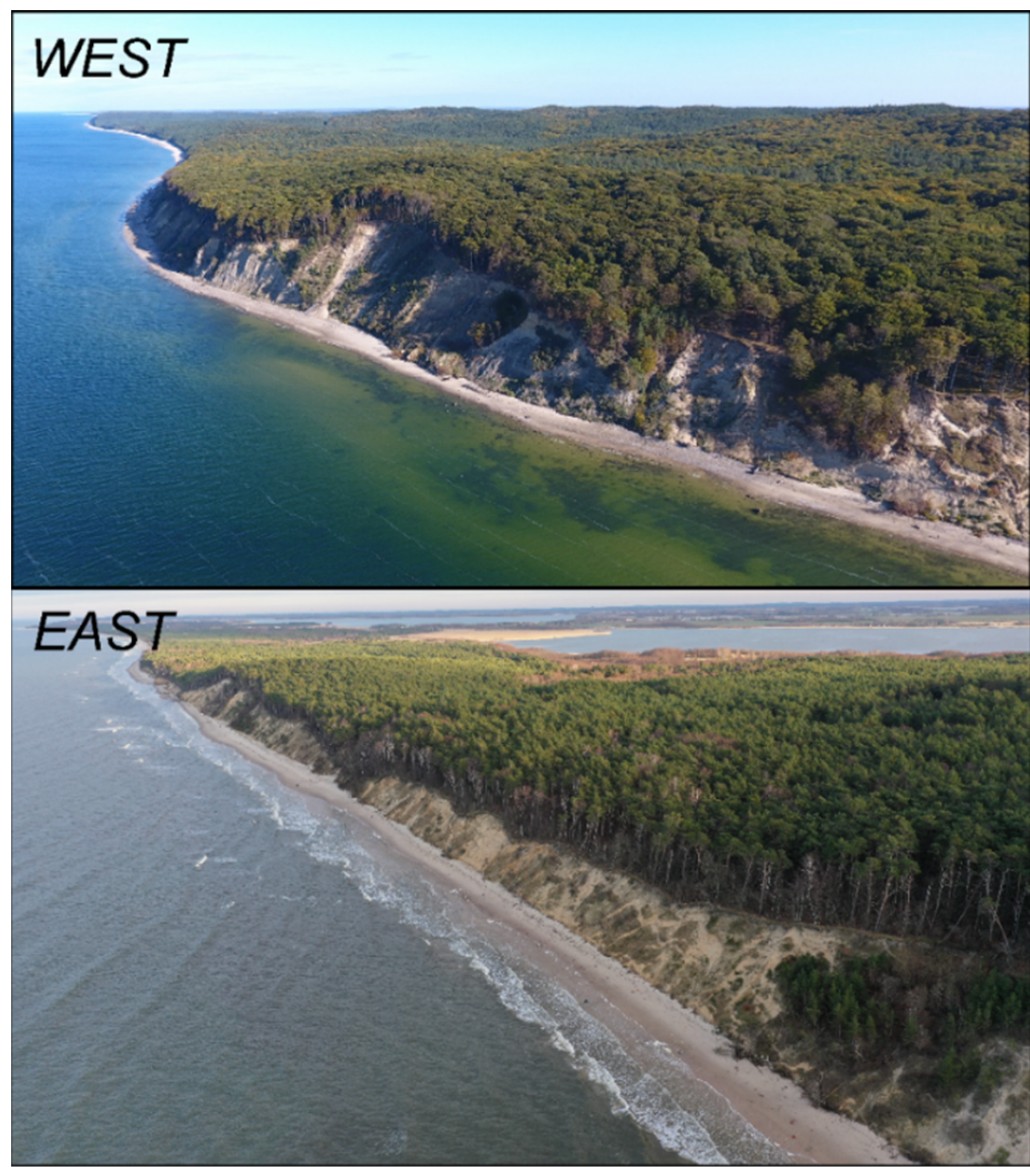

**Figure 2.** Morphological and lithological variability of western and eastern section of Wolin Island cliff coast. The upper image presents active high cliffs made of clay and sandy clay sediments with well-developed landslides. The bottom image presents active sandy cliffs moderately covered with grass. Oblique image taken by a drone (DJI Mavic 2 Pro) at an altitude of 100 m (Source: this study).

**Table 1.** Survey dates and basic parameters of analyzed point clouds (Source: this study).

| Parameter | 2012 | 2015 | 2018 | 2020 |
|---|---|---|---|---|
| Number of points (all classes) | 216,253,380 | 437,275,340 | 431,853,371 | 492,283,200 |
| Number of points (class 2: ground) | 24,152,629 | 100,294,232 | 77,201,988 | 91,718,059 |
| Percent of ground points (%) | 11.17 | 22.94 | 17.88 | 16.63 |
| Point cloud density of ground points (pts/m$^2$) | 3 | 7 | 6 | 8 |
| Survey date | 20 September 2012 | 26 September 2015 | 9 September 2018 | 20 October 2020 |

Sea hydrodynamics data included hourly values of sea level, wave height, and its azimuth from the period of 2012–2020. Sea level data come from the mareograph installed in Świnoujście and were made available by the Institute of Meteorology and Water Management. On the basis of these data, the maximum, average, and minimum sea level values were determined in the three analyzed research periods (2012–2015, 2015–2018, and 2018–2020). In case of the Polish coast, the elevation of the zero gauge as a reference is 500 cm N.N. (Normal Null—the Amsterdam zero in NAP Normal Amsterdam Peil, EVRS European Vertical Reference System). The article presents sea levels N.N., where the zero level N.N. is equal to 500 cm on the gauges [66]. In the Polish coastal zone of the Baltic Sea, a sea level $\geq$ 70 cm NN is assumed as a storm surge [67]. The study presents the temporal distribution of sea level occurrence, which played a particularly important role in the geomorphological changes of the cliff coast—i.e., the level generating beach erosion $\geq$60 cm, and especially the level eroding the cliff slope $\geq$90 cm [68].

The wave altitude and azimuth data come from the WAve Model (WAM) [69], which was adapted for the Baltic Sea by [70] and is made available by the Interdisciplinary Modeling Center of the University of Warsaw. The wave height data made it possible to determine the energy of storm surges L [71], in accordance with the formula

$$L = \Sigma\left(T \times H_t{}^2\right),$$

where T [h]—time of sea level (for $T_b$ $H_{max} \geq 60$ cm N.N.—beach erosion sea level) and (for $T_c$ $H_{max} \geq 90$ cm N.N.—cliff erosion sea level), $H_t$ [m]—wave height (average of 1/3 of the largest waves in a given period).

The direction of the wave $W_d$ [º] is also presented, which shows the exposure of the shoreline particularly exposed to erosive processes.

*2.3. GIS Analysis*

The acquired light detection and ranging (LiDAR) data were the basis for the implementation of digital elevation models (DEM), on the basis of which the DoD (DEM of Difference) step-by-step models were created. Based on the obtained differential models, the volume changes of the cliff, occurring in the assumed time intervals (2012–2015, 2015–2018, 2018–2020, and 2012–2020), were calculated. The analyses carried out included positive changes (which we interpret as deposition) and negative changes (erosion). The total value of the obtained results was the final budget of the sediments.

In the first stage of the analyses, on the basis of the obtained point clouds for all years (2012, 2015, 2018, 2020), digital elevation models (DEM) were made. The models were built on the basis of the value of class 2 (ground) points. DEM models were made in the ArcGis program (conversion tool: Las dataset to raster) in the coordinate system (ETRS_1989_Poland_CS92). The point cloud density analysis (Table 1) was the basis for determining the raster resolution. It is assumed that at least 2 points are required to determine the cell value, assuming that more of them increases the accuracy of the cell value. Finally, it was decided to build a raster with a resolution of 1 m. The binning interpolation method was used to determine the cell value, taking into account the mean value of the points lying within it, which was the optimal solution in the case of the analysis of steep slopes. Using a minimum value on steep slopes could underestimate the actual height.

Due to the considerable length of the coast, it was divided into two equal parts: west and east, 4.6 km each. In order to determine the spatial diversity of changes taking place on the cliffs, the analyzed coastal fragments were divided into equal sectors with a length of 200 m. In total, 48 sectors were obtained (Figure 3). The research covered only the cliff slope, i.e., the area stretching from the top of the cliff to the foot of the cliff with colluvial forms.

The knowledge of the accuracy of the raster cell values was very important for the volumetric analysis. This information was important to determine the minimum level of detection in later volumetric analyses. For this purpose, stable objects (whose position and

height did not change over time) were selected as reference points on raster models. The elevation root mean square error of the reference points determined the minimum level of detection. In total, 28 reference points were selected for the analysis (15 points on the western section and 13 points on the eastern section) (Figure 3). Elevation RMSE analyses were determined separately for the eastern and western parts in pairs for the years covered by the analysis (2012–2015, 2015–2018, 2018–2020, 2012–2020). The obtained values were within the limits of W: 0.05–0.1 m and E: 0.07–0.1 m (Table 2).

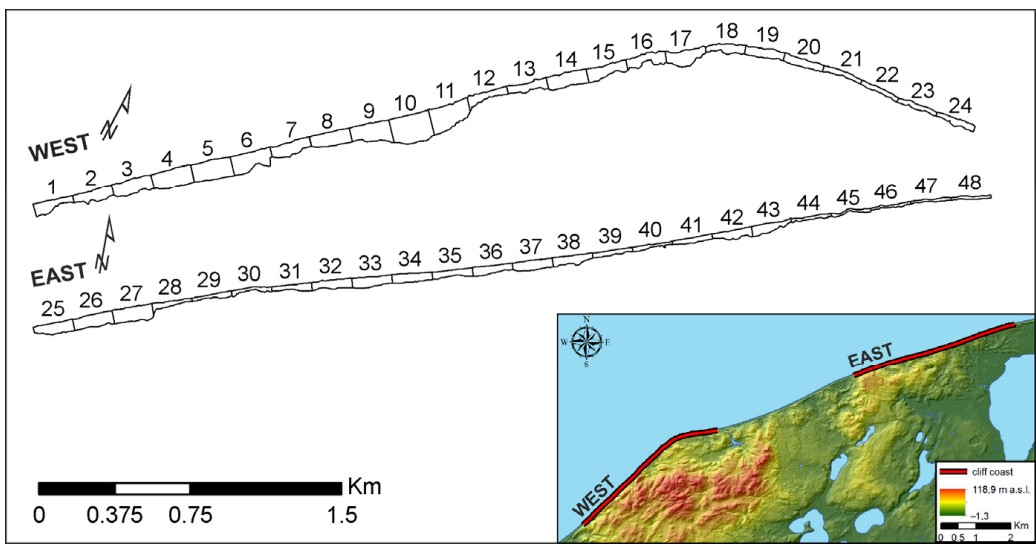

**Figure 3.** Mask of interest showing the analyzed cliff area. The lower map shows the analyzed western and eastern cliff sections and the distribution of reference points (Source: this study).

The differential analysis of the DTM models was performed using the GCD (geomorphic change detection) application [72–74]. The GCD application is an ArcGIS plug-in used to analyze the morphological variability of various types of relief, e.g., river beds, mountain slopes, and sea coasts, including cliffs. Due to the automation of analyses, differential models (DoD) were built. The software, taking the measurement error into account, counts the statistics of the volumetric changes of the landform and determines the uncertainty range of the obtained results.

**Table 2.** Elevation root mean square errors calculated for pairs of DTMs, which were taken into consideration for minimum level of detection estimation (Source: this study).

| DoD | West Section RMSE (m) | East Section RMSE (m) |
|---|---|---|
| 2012–2015 | 0.07 | 0.1 |
| 2015–2018 | 0.1 | 0.1 |
| 2018–2020 | 0.07 | 0.07 |
| 2012–2020 | 0.05 | 0.07 |

In the first stage of the analyses, it was necessary to import DEM models to the geomorphic change detection (GCD) program. Then, the models were subjected to differential analysis, indicating the simple minimum level of detection based on an earlier analysis of reference points. The calculated volume differences were limited to the range of the previously prepared AOI (area of interest) masks (range from the cliff top line to the cliff foot line along the entire length of the analyzed coast) (Figure 3). Due to the fact that the surface of the slope was different in the subsequent observation periods, AOI masks were prepared for each of the performed differential analyses (4 in total). Then each mask was divided into 200 m long sectors and used for budget segregation in order to capture the spatial variability of volume changes.

As a result, the total net volume difference was calculated in $m^3$ and its two components: erosion (total volume of surface lowering) and deposition treated as an accumulation of sediments (total volume of surface rising). Additionally, when indicating the minimum level of detection, the volume error and its percentage share are calculated for each of the calculated indicators. The results are visualized in the form of differential models indicating the spatial differentiation of erosion and deposition zones. Surfaces with values in the uncertainty range are assigned the NoData value.

The obtained results are presented in the form of the total budget ($T_B$), which presents the difference in the volume of deposited (+) and eroded (−) sediments along the entire length of the investigated coast, according to the formula

$$T_B = E_V - D_V$$

where $T_B$—total budget, $E_V$—eroded volume, and $D_V$—deposited volume.

In order to spatially normalize the obtained budget values, it was proposed to use two simple coastal erosion indicators. The first is the length-normalized sediment budget ($L_B$), which shows the budget of sediments per one meter of coast length ($m^3/m$). This is quite significant information that points to the efficiency of the cliff. The eroded sediments are delivered to the nearshore area and participate in the longitudinal transport. The quantity of the dynamic layer in the nearshore is valuable information in predicting the morphodynamics of the entire coastal system (nearshore, beach, cliff).

$$L_B = T_B/L$$

where $L_B$—length-normalized sediment budget, $T_B$—total budget, and L—cliff length.

Another indicator specifies the volume of eroded sediments per unit area of the cliff, i.e., area-normalized sediment budget, (AB), and is expressed in $m^3/m^2$. This value represents the amount of sediment that has been eroded per cliff surface. This indicator actually represents the degree of erosion of the cliff. This information, in turn, is important in the context of assessing the degree of activity of the cliff.

$$A_B = T_B/A$$

where $A_B$—area-normalized sediment budget, $T_B$—total budget, A—cliff area.

The presented indicators are a proposal for the standardization of the results of the coastal erosion volumetric analysis. Due to the use of spatially and temporally normalized indicators in research, it is possible to compare results from various regions of the world.

Due to the fact that the calculated indicators related to multi-year periods, for the time normalization they were converted to one-year intervals $L_B$ ($m^3/m/a$), $A_B$ ($m^3/m^2/a$).

In order to determine the dynamics of the Wolin Island cliff coast, the obtained erosion indexes were assigned erosion intensity classes. The categorization consisted of assigning four equal classes with regard to the range of obtained values of erosion indicators ($L_B$, $A_B$) from each sector (Table 3).

**Table 3.** Sediment loss categories (Source: this study).

| The Intensity of Erosion | $L_B$ ($m^3/m/a$) | $A_B$ ($m^3/m^2/a$) |
|:---:|:---:|:---:|
| Low | 1.00–5.25 | 0–0.125 |
| Average | 5.26–9.50 | 0.126–0.250 |
| High | 9.51–13.75 | 0.251–0.375 |
| Very high | 13.76–18.00 | 0.376–0.500 |

## 3. Results

### 3.1. Hydrodynamic Conditions of Coastal Erosion

In the period 2012–2020, the mean sea level was 6 cm N.N. and increased from 3 cm N.N. in the first research period up to 9 cm N.N. in the third research period (Table 4). Currently, the southern Baltic Sea level is increasing by about 3–4 mm/a. The relatively high frequency of days with erosive sea level $H_{max} \geq 90$ cm occurred especially in the second research period 2015–2018 (Figure 3). At that time, five such storm surges were recorded, which lasted a total of 80 h. The highest sea level was 140 cm N.N. during the event on 4–5 January 2017 (Figure 4). In the remaining research periods, erosive storm surges were 3–4 times less and they were less dynamic. Especially in the first research period, September 2012–September 2015, when only two such cases were recorded. Total duration of the maximum sea level ≥90 cm N.N. was only 17 h (Table 5).

**Table 4.** Parameters of sea level in measurements period. Raw data source—Institute of Meteorology and Water Management.

| Period | Date | Sea Level (cm) N.N. Reference Level | | | Number of Days | |
|---|---|---|---|---|---|---|
| | | $H_{average}$ | $H_{max}$ | $H_{min}$ | Beach Erosion Sea Level $H_{max} \geq 60$ cm | Cliff Erosion Sea Level $H_{max} \geq 90$ cm |
| First | 28 September 2012 –26 September 2015 | 3 | 100 | −106 | 21 | 3 |
| Second | 27 September 2015 –9 September 2018 | 8 | 140 | −92 | 32 | 7 |
| Third | 10 September 2018 –20 October 2020 | 9 | 133 | −126 | 30 | 2 |
| Total | 28 September 2012 –20 October 2020 | 6 | 140 | −126 | 83 | 12 |

In the second measurement period, potentially the most favorable sea conditions of increased erosion of the cliff coast occurred. The reverse hydrodynamic situation occurred in the first measurement period.

A characteristic feature of erosive storm surges with $H_{max} \geq 90$ cm was their rare occurrence—only nine cases in the analyzed 8 years. The statistical repeatability of erosive storm surges was only one event per year. There have been years when such events did not occur at all (2014 and 2018). There have also been years of increased turnout, e.g., 3 cases in 2017. Such clustering of events poses a particular threat to the geomorphological stability of the seashore. The duration of the analyzed storm surges was short, usually around 33 h (from 17 to 47 h), of which the erosive sea level lasted from 1 h to 32 h. The rate of water level rise was even over 20 cm/h. The wave height during erosive storm surges was from 2.3 m to 4.0 m outside the arrival zone (Table 5). Very often, the highest wave heights were before the peak of sea level (Figure 5). The direction of the wave was conducive to the cliff coast erosion, as it was most often 'perpendicular' to the dominant NW-NE exposure of the shoreline. The energy of storm surges varied in particular research periods. Cumulatively, the highest value of storm energy (L) with $H_{max} \geq 90$ cm occurred during floods in the second measurement period and it was L = 707. Particularly strong erosive storm surges with $H_{max} \geq 90$ cm occurred on January 5, 2017 (L = 252) and on 29–30 October 2017 (L = 274). On the other hand, the lowest energy of storm surges occurred in the first measurement period and was for two storm events with $H_{max} \geq 90$ cm only L = 137 (Table 5 and Figure 5).

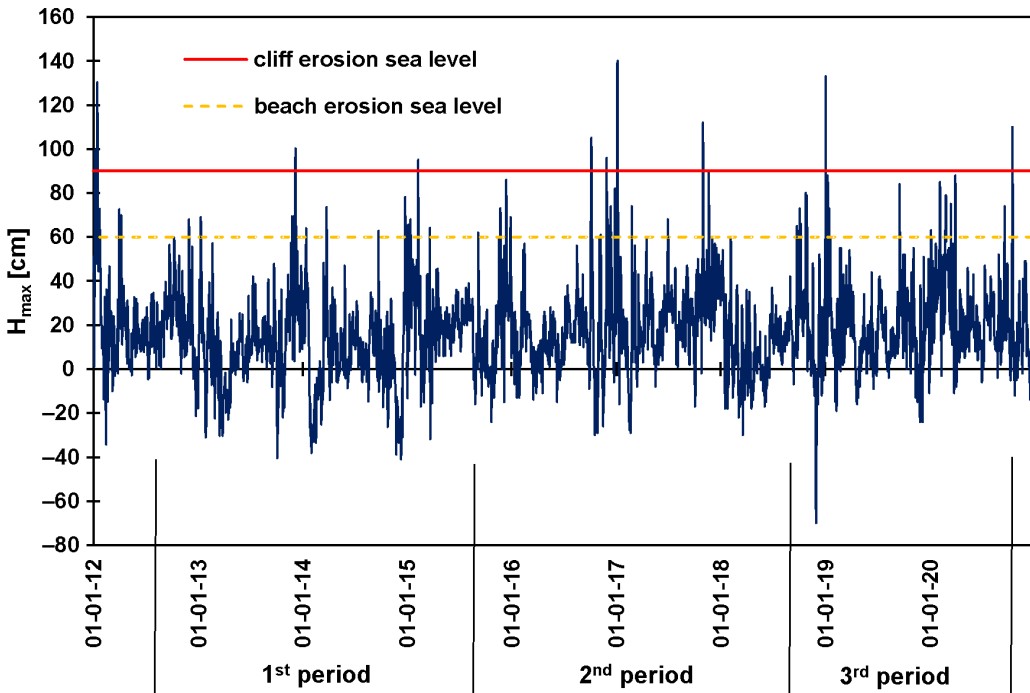

**Figure 4.** Maximum sea level in the research period (Świnoujście—N.N. reference level). Raw data source—Institute of Meteorology and Water Management.

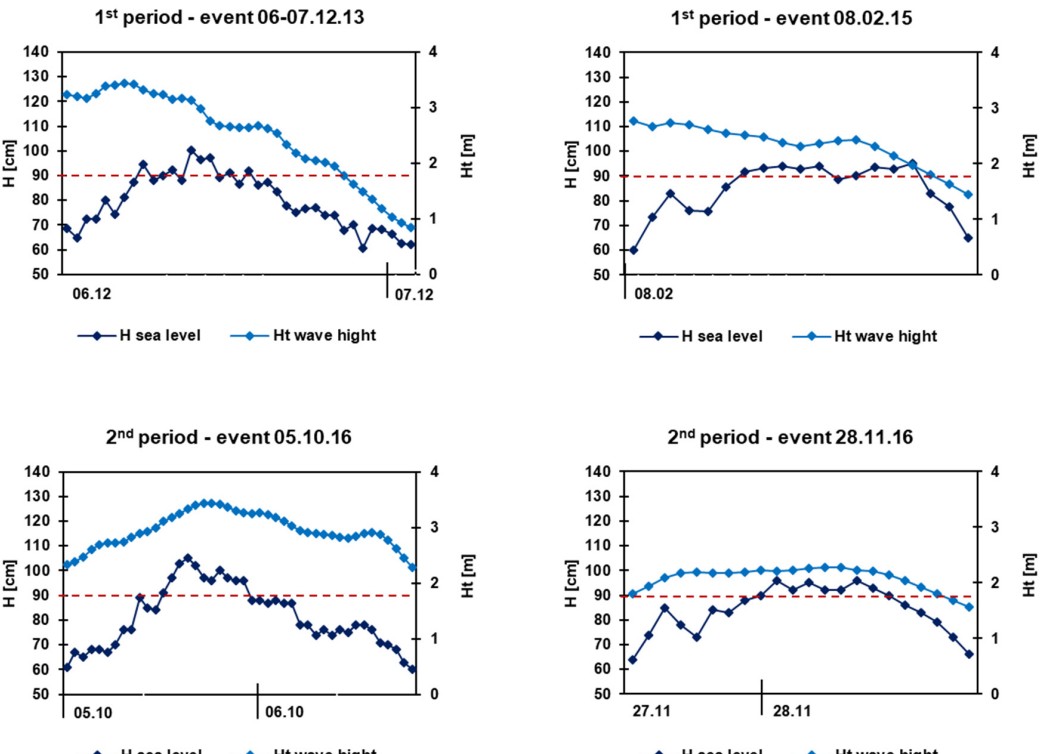

**Figure 5.** *Cont.*

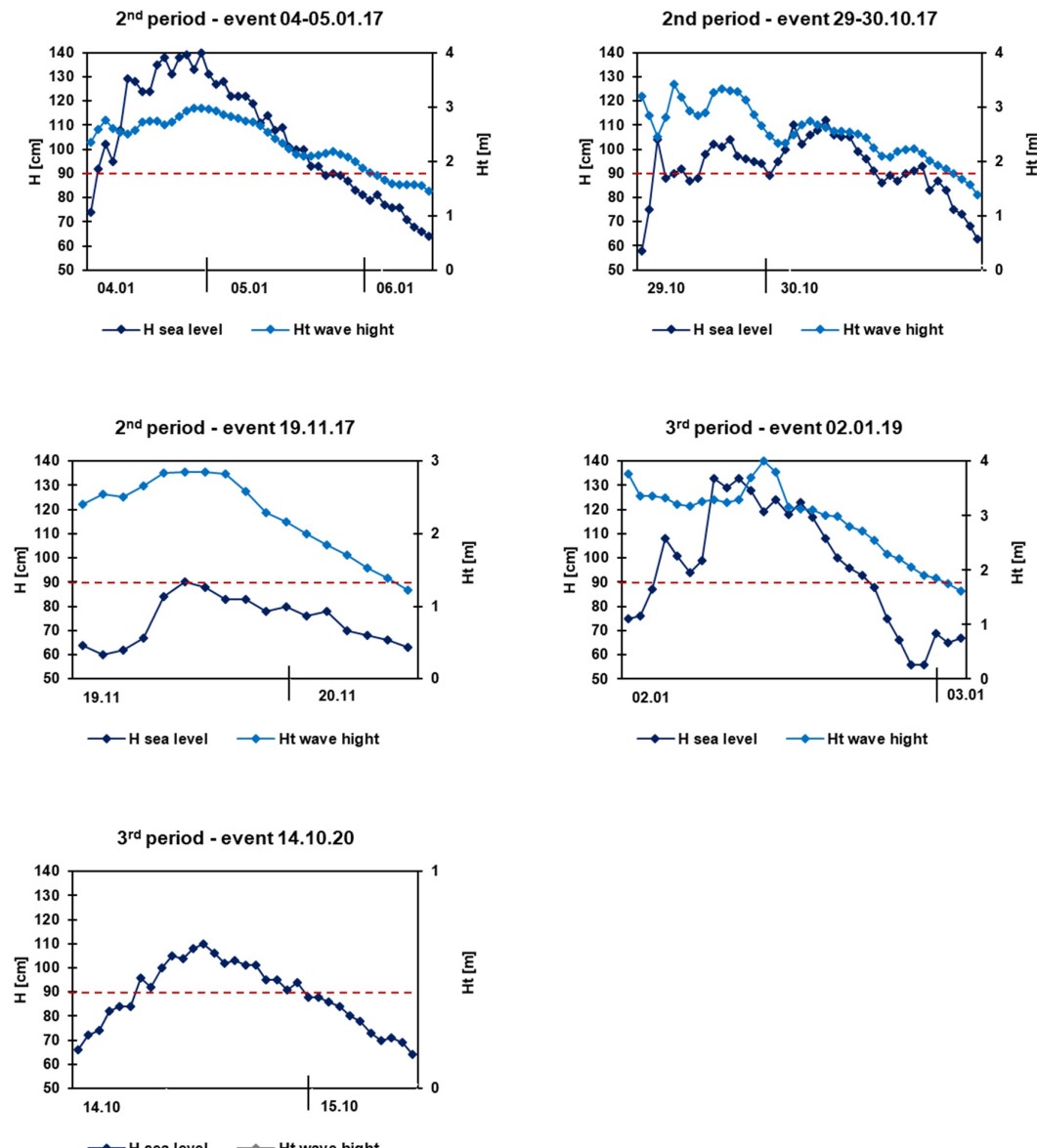

**Figure 5.** Sea hydrodynamics parameters (maximum sea level and high of wave) during storm surges on $H_{max} \geq 90$ cm N.N. (2012–2020 Świnoujście). Raw data source—Institute of Meteorology and Water Management, Interdisciplinary Modeling Center of the University of Warsaw.

**Table 5.** Parameters of hydrodynamic properties (Świnoujście—N.N. reference level). Raw data source—Institute of Meteorology and Water Management, Interdisciplinary Modeling Center of the University of Warsaw.

| Period | Number of Events | Time of Duration (h) | Data of Storm Surges | $H_{max}$ Sea Level Maximum (cm) | $H_{ave60}$ Sea Level Average by $H_{max} \geq 90$ cm (cm) | $H_{ave90}$ Sea Level Average by $H_{max} \geq 60$ cm (cm) | $T_{90}$ Time of Duration by $H_{max} \geq 90$ cm (h) | $T_{60}$ Time of Beach Erosion by $H_{max} \geq 60$ cm (h) | $H_{r90}$ Maximum Sea Level Rise by $H_{max} \geq 90$ cm (cm/h) | $H_{r60}$ Maximum Sea Level Rise by $H_{max} \geq 60$ cm (cm/h) | $H_{f90}$ Maximum/Considerable Wave Hight in Sea by $H_{max} \geq 90$ cm (m) | $H_{f60}$ Maximum/Considerable Wave Hight in Sea by $H_{max} \geq 60$ cm (m) | $W_d$ Wave Direction/Shoreline Exposure (°) | $L_b$ Storm Energy by $H_{max} \geq 60$ cm (Beach Erosion) | $L_c$ Storm Energy by $H_{max} \geq 90$ cm (Cliff Erosion) |
|---|---|---|---|---|---|---|---|---|---|---|---|---|---|---|---|
| 28 September 2012–26 September 2015 | 2 | 17 | 6–7 December 2013 | 100 | 94 | 80 | 8 | 37 | 12 | 8 | 3.3/3.2 | 3.4/3.2 | 143/NW | 390 | 84 |
| | | | 8 February 2015 | 95 | 93 | 85 | 9 | 19 | 6 | 14 | 2.5/2.4 | 2.8/2.6 | 169/N | 131 | 53 |
| 27 September 2015–9 September 2018 | 5 | 80 | 5 October 2016 | 105 | 98 | 81 | 11 | 44 | 7 | 13 | 3.4/3.4 | 3.4/3.3 | 214/NE | 481 | 128 |
| | | | 28 November 2016 | 96 | 93 | 84 | 9 | 22 | 6 | 11 | 2.3/2.2 | 2.3/2.2 | 189/N | 108 | 45 |
| | | | 4–5 January 2017 | 140 | 117 | 104 | 32 | 47 | 21 | 2 | 3.0/2.8 | 3.0/2.8 | 196/N | 370 | 252 |
| | | | 29–30 October 2017 | 112 | 99 | 92 | 27 | 43 | 29 | 17 | 3.4/3.2 | 3.4/3.1 | 161/N | 408 | 274 |
| | | | 19 November 2017 | 190 | 90 | 74 | 1 | 17 | 6 | 17 | 2.9/2.9 | 2.9/2.8 | 148/NW | 130 | 8 |
| 10 September 2018–20 October 2020 | 2 | 33 | 2 January 2019 | 133 | 113 | 97 | 17 | 28 | 34 | 13 | 4.0/3.5 | 4.0/3.5 | 195/N | 342 | 207 |
| | | | 14 October 2020 | 110 | 100 | 88 | 16 | 33 | 12 | 8 | - | - | - | - | - |
| 28 September 2012–20 October 2020 | Total | | Maximum | | | | | | | | | | | | |
| | 9 | 130 | | 140 | 100 | 87 | 32 | 47 | 29 | 17 | 4.0/3.5 | 4.0/3.5 | | 481 | 274 |

### 3.2. Volume Changes of the Cliff Coast

The analysis of sea levels showed that in the first period (2012–2015) there were only two erosive storm surges. At that time, the low dynamics of the sea was reflected in the low activity of geomorphological changes of the cliff coast. In the analyzed period, $T_B = 79{,}904\ m^3 \pm 20{,}541\ m^3$ sediments were removed from the cliff (Figure 6), which accounted for only about 16% of eroded sediments in the entire observation period (2012–2020). The annual value per 1 m of coast was $L_B = -2.8 \pm 0.8\ m^3/m/a$, and the surface ratio $A_B = -0.06 \pm 0.02\ m^3/m^2/a$ (Table 6). The greatest volume changes of the cliff occurred within sectors: 34 ($L_B = -10.5 \pm 1.1\ m^3/m/a$; $A_B = -0.27 \pm 0.03\ m^3/m^2/a$), 19 ($L_B = -9.8 \pm 0.8\ m^3/m/a$, $A_B = -0.24 \pm 0.02\ m^3/m^2/a$), and 32 ($L_B = -9.2 \pm 1.0\ m^3/m/a$; $A_B = -0.25 \pm 1.22\ m^3/m^2/a$) (Figures 7–9). In the case of Sectors 32 and 34, a large defect in the cliff was conditioned by a significant share of near-surface sandy sediments. This sediment series is characterized by the lowest resistance to erosive processes. An exceptional situation occurred in the case of Sector 19, where erosion-resistant clay sediments predominate in the construction of the cliff. During the first storm surge (July 2013), these sediments were quite clearly undercut, as a result of which the entire upper (sandy) part of the cliff collapsed. Extensive colluvial covers, not very resistant to erosion, have been accumulated on the beach. During subsequent storm surges (e.g., February 2015), the colluvium covers were mostly eroded. In turn, the lowest dynamics of the cliff was recorded in the western part of the research area in Sectors 1–9, where material accumulation dominated (average $L_B = 0.99 \pm 0.9\ m^3/m/a$; $A_B = 0.02 \pm 0.01\ m^3/m^2/a$). Accumulation was due to the redeposition of sediments from the top of the cliff. The volume of accumulated material was very often within the uncertainty range.

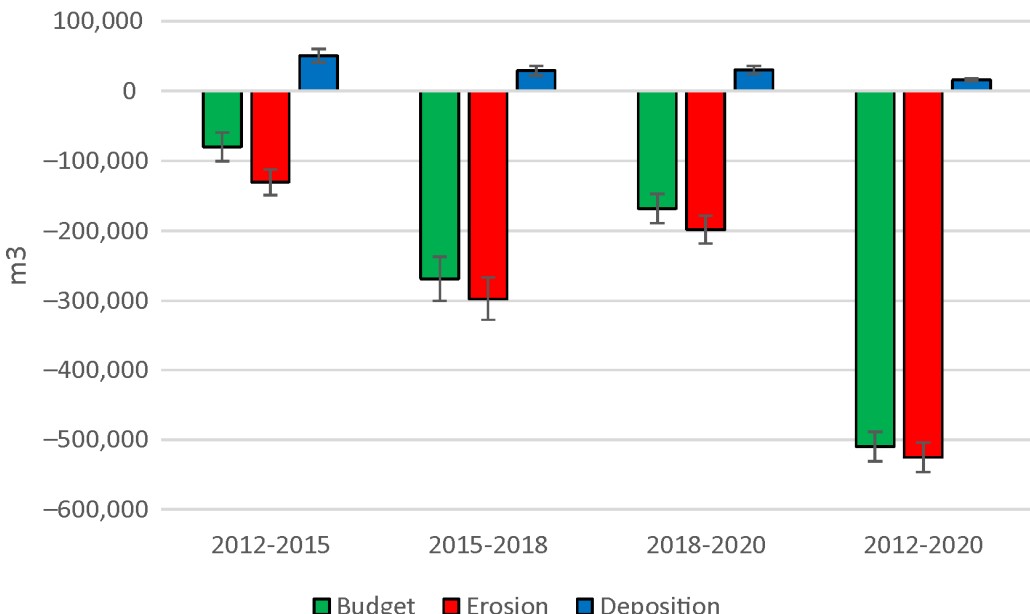

**Figure 6.** Volume changes of the cliff in the analyzed time intervals. Raw data source—Maritime Office in Szczecin.

In the second research period (2015–2018), the sea dynamics was greater, as evidenced by the occurrence of 5 erosive storm surges. An extreme event was the crossing of the deep Axel low over the Baltic Sea (4 January 2017), which caused an extreme storm surge (+140 cm N.N.). This event greatly influenced the volume changes of the cliffs. The impact of sea waves on the cliff in the analyzed period resulted in a loss of sediment by volume $T_B = 268{,}346 \pm 31{,}384\ m^3$ ($L_B = 9.3 \pm 1.1\ m^3/m/a$; $A_B = 0.21 \pm 0.03\ m^3/m^2/a$), (Figure 6). These were losses more than three times greater than in the years 2012–2015. In the period 2015–2018, the loss of sediments was 53% in relation to the entire eight-year research period. The greatest loss of sediment was recorded in Sectors 32–35 (on average $L_B = -25.1 \pm 1.3\ m^3/m/a$; $A_B = -0.6 \pm 0.03\ m^3/m^2/a$) and 10 ($L_B = -25.2 \pm 3.6\ m^3/m/a$; $A_B = -0.19 \pm 0.01\ m^3/m^2/a$)

(Figures 7–9). The parts of the cliffs where the greatest erosion has been observed are mostly composed of sand sediments. The lowest dynamics of cliffs was observed within sectors with clay structure: 20 ($L_B = -0.1 \pm 0.7$ m³/m/a; $A_B = -0.002 \pm 0.02$ m³/m²/a) and 23 ($L_B = -0.2 \pm 0.4$ m³/m/a; $A_B = -0.01 \pm 0.02$ m³/m²/a).

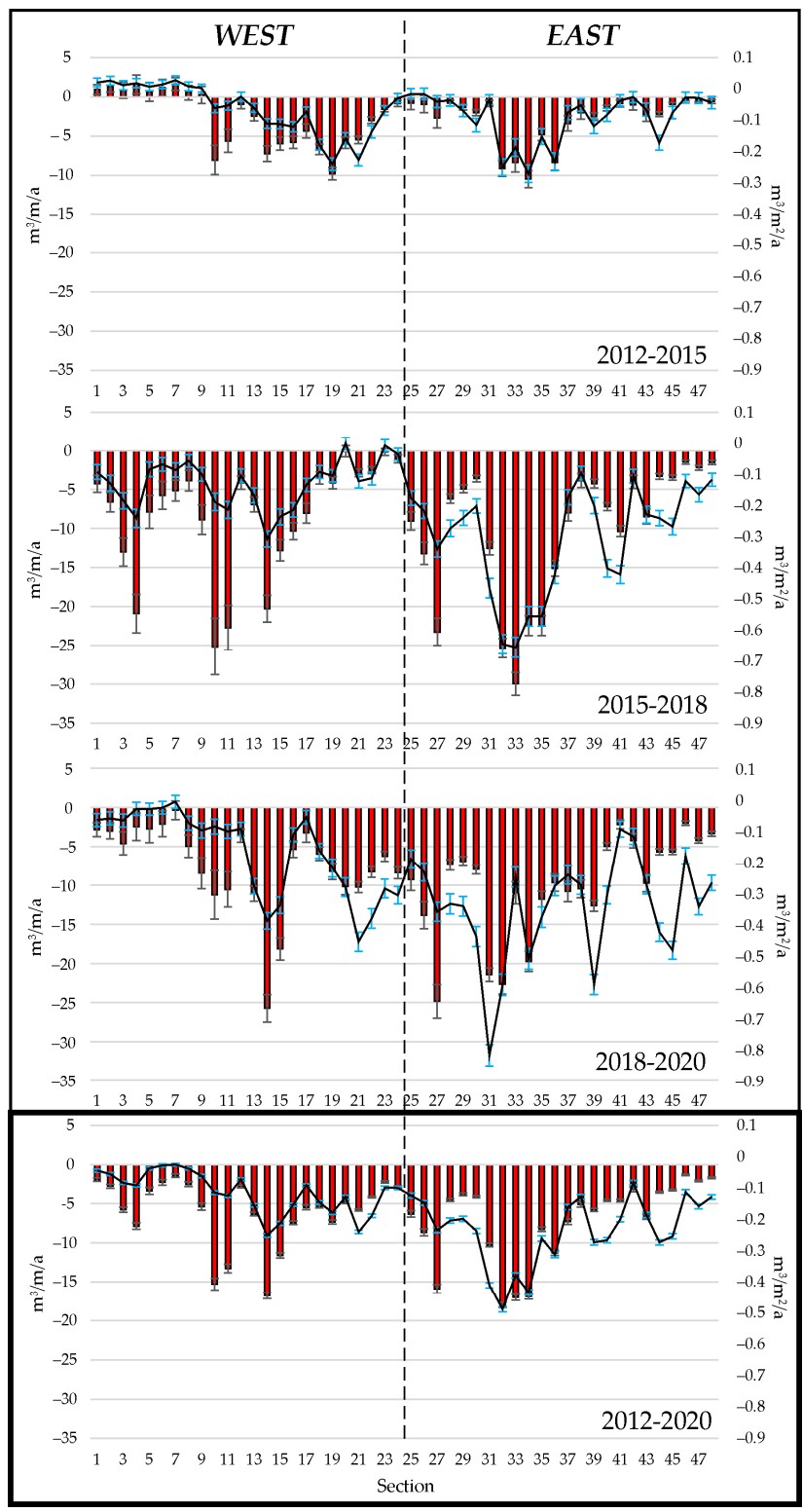

**Figure 7.** Spatial differentiation of the budget of sediments in individual sectors in the analyzed time periods. Raw data source—Maritime Office in Szczecin.

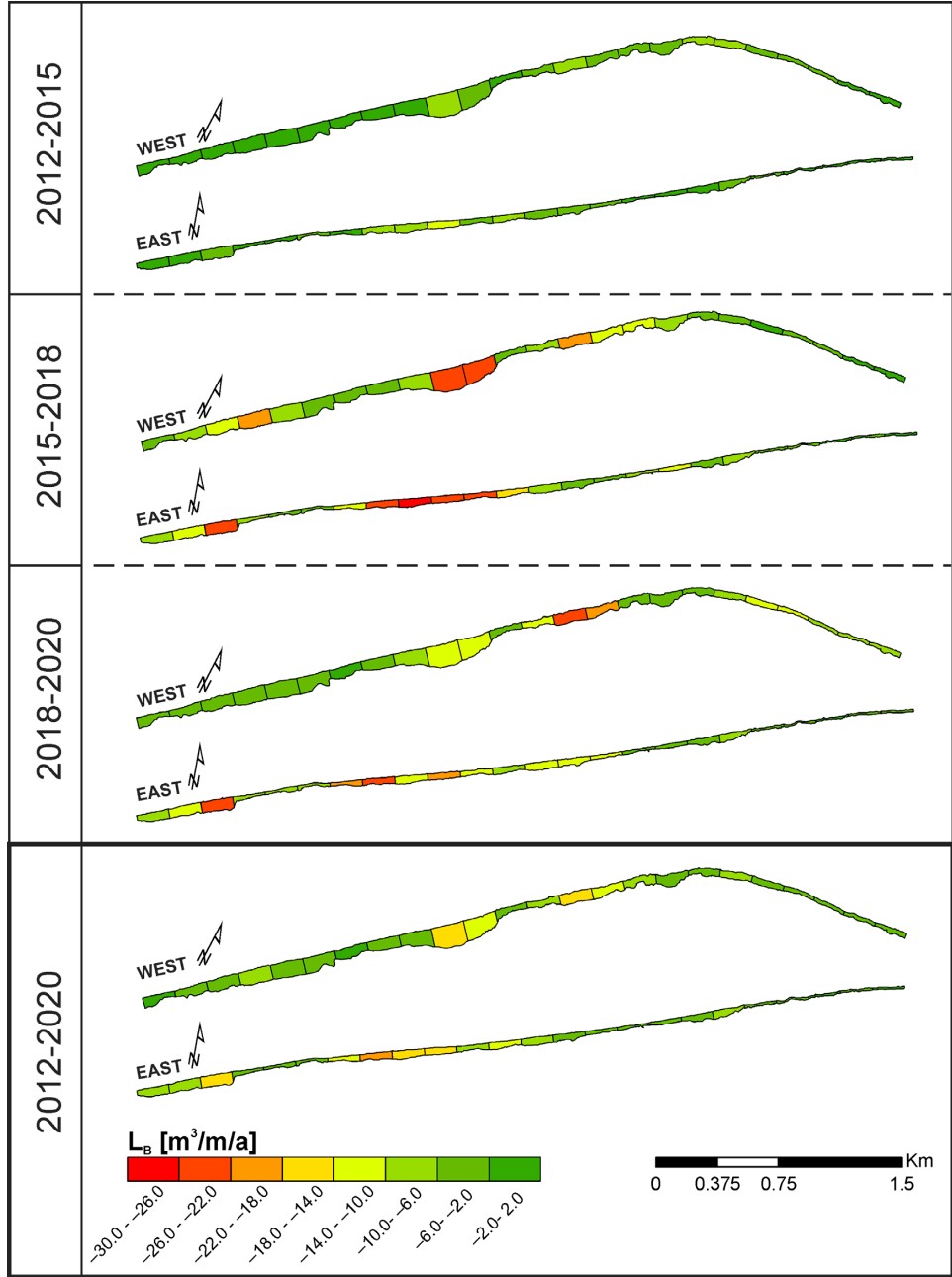

**Figure 8.** Figure showing the spatial distribution of the $L_B$ index in analyzed sectors in all research periods. Raw data source—Maritime Office in Szczecin.

In the last of the analyzed periods (2018–2020), the sea dynamics were slightly lower than in the previous period. There were two storm events during that time, as a result of which the cliff coast was significantly transformed. It should be noted that the last research period was the shortest and lasted only 2 years. The total amount of material discharged into the sea was $T_B = 168,090 \pm 20,783$ m$^3$, which accounted for 31% of eroded sediments in 2012–2020 (Figure 6). The annual sediment budget indicators showed slightly lower cliff dynamics than in the second period ($L_B = -8.9 \pm 1.0$ m$^3$/m/a; $A_B = -0.26 \pm 0.03$ m$^3$/m$^2$/a). The highest dynamics was observed within the sandy cliffs: Sector 14 ($L_B = -25.7 \pm 1.8$ m$^3$/m/a; $A_B = -0.39 \pm 0.03$ m$^3$/m$^2$/a), Sector 27 ($L_B = -24.8 \pm 2.1$ m$^3$/m/a; $A_B = -0.36 \pm 0.03$ m$^3$/m$^2$/a), Sector 32 ($L_B = -22.6 \pm 1.3$ m$^3$/m/a; $A_B = -0.59 \pm 0.03$ m$^3$/m$^2$/a), and Sector 31 ($L_B = -21.4 \pm 0.9$ m$^3$/m/a; $A_B = -0.82 \pm 0.03$ m$^3$/m$^2$/a). The lowest activity of erosion processes was observed on

the western edge of the studied coast in Sectors 1–7 (on average $L_B = -2.6 \pm 1.4$ m$^3$/m/a; $A_B = -0.04 \pm 0.02$ m$^3$/m$^2$/a). Low dynamics were also observed in the eastern coastal zone in Sectors 46–48 (on average $L_B = -3.2 \pm 0.3$ m$^3$/m/a; $A_B = -0.26 \pm 0.03$ m$^3$/m$^2$/a), which is characterized by a fairly significant retraction of the cliff foot inland. In this case, the wide beach is quite a good protection of the shore against storm surges. Due to the low height of the cliff occurring here, despite the negligible total erosion of the shore, the surface ratio of the budget is relatively high, e.g., in the Sector 47 ($A_B = -0.34 \pm 0.03$ m$^3$/m$^2$/a), (Figures 7–9).

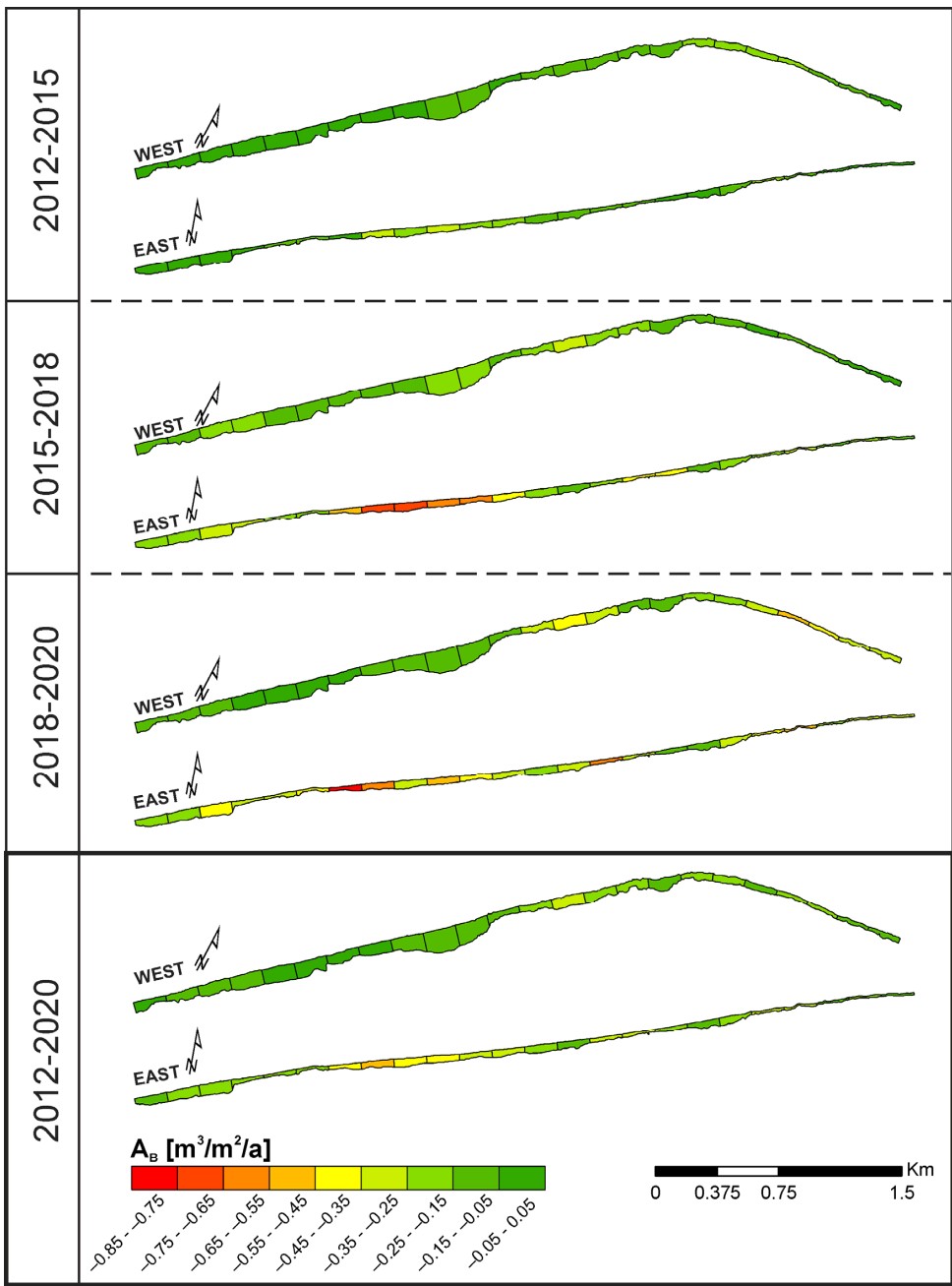

**Figure 9.** Figure showing the spatial distribution of the $A_B$ index in analyzed sectors in all research periods. Raw data source—Maritime Office in Szczecin.

The measurement period analyzed in the article covered 8 years (2012–2020), divided into three periods (2012–2015, 2015–2018, and 2018–2020). In total, from the almost-10-km section of the cliff coast, it was discharged into the sea within 8 years $T_B = 509{,}909 \pm 21{,}111$ m$^3$

($L_B = 6.6 \pm 0.3$ m$^3$/m/a; $A_B = 0.17 \pm 0.01$ m$^3$/m$^2$/a). This budget consisted of $E_V = 525,513 \pm 21,003$ m$^3$ eroded sediments and $D_V = 15,603 \pm 2130$ m$^3$ accumulated sediments redeposited from the top of the cliff. It should be emphasized that the obtained values of erosion and accumulation (in the range of 2012–2020) are not total values from individual time intervals, but are the result of differentiating the height models from 2012 and 2020. Such an analysis made it possible to obtain results with a lower error than in the case of summing up the values from individual measurement periods, in the case of which the analytical errors would be cumulated. The conducted analyses show that the dynamics of cliffs in the analyzed period was conditioned mainly by the geological structure. Sandy cliffs in the eastern part of the research area in Sectors 32–34 (on average $L_B = 17.3 \pm 0.4$ m$^3$/m/a; $A_B = 0.43 \pm 0.01$ m$^3$/m$^2$/a) and in western part in Sector 14 ($L_B = 16.8 \pm 0.4$ m$^3$/m/a; $A_B = 0.25 \pm 0.01$ m$^3$/m$^2$/a) (Figures 7–9).

Low and medium erosion (Table 3, Figure 10) occurred mainly on the sections of stabilized cliffs, densely covered with forest, as well as on cliffs built of clay sediments more resistant to erosion processes. Such cases occurred in the extreme parts of the western and eastern parts of the coast in Sectors 1–8, 16–26, 28–30, and 37–48. In turn, high and very-high erosion occurred on cliffs built of sandy sediments that are not very resistant to erosive processes. It should be emphasized that, in the western part of the analyzed research area, the total length of the coast characterized by high and very high erosion was 0.8 km, while in the eastern part it was 50% greater (1.2 km). In the case of the area indicator ($A_B$), which apart from the absolute budget also takes into account the cliff surface, high and very-high erosion in the western part did not occur in any sector, while in the eastern part there were 10 such cases (2.0 km of shore). The above situation results directly from the large variation in the height of the analyzed cliffs (Figure 10). In the western part, there are clearly higher cliffs than in the eastern part. The above analysis shows that in the case of high cliffs, despite a large loss of material, the surface erosion indicator ($A_B$), which indicates the degree of cliff erosion, may often be lower than in the case of low cliffs showing a relatively small loss of sediment. This relationship is very well visualized on the cliff profiles in 2012 and 2020 (Figure 11).

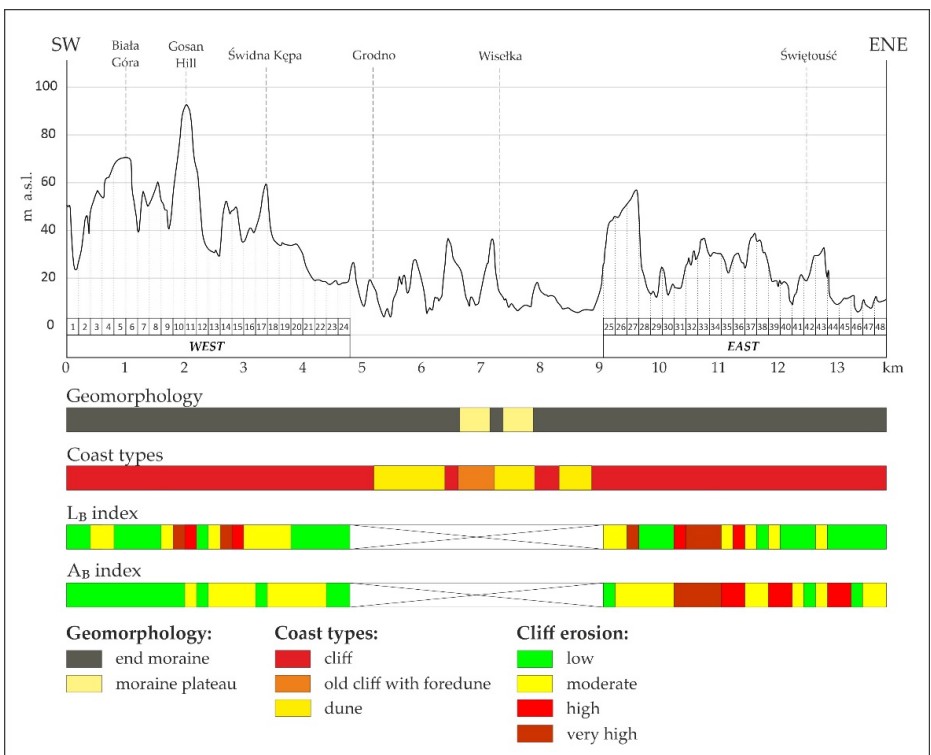

**Figure 10.** Spatial diversity of cliff erosion (in terms) on the background of coastal geomorphology and typology. Raw data source—Maritime Office in Szczecin.

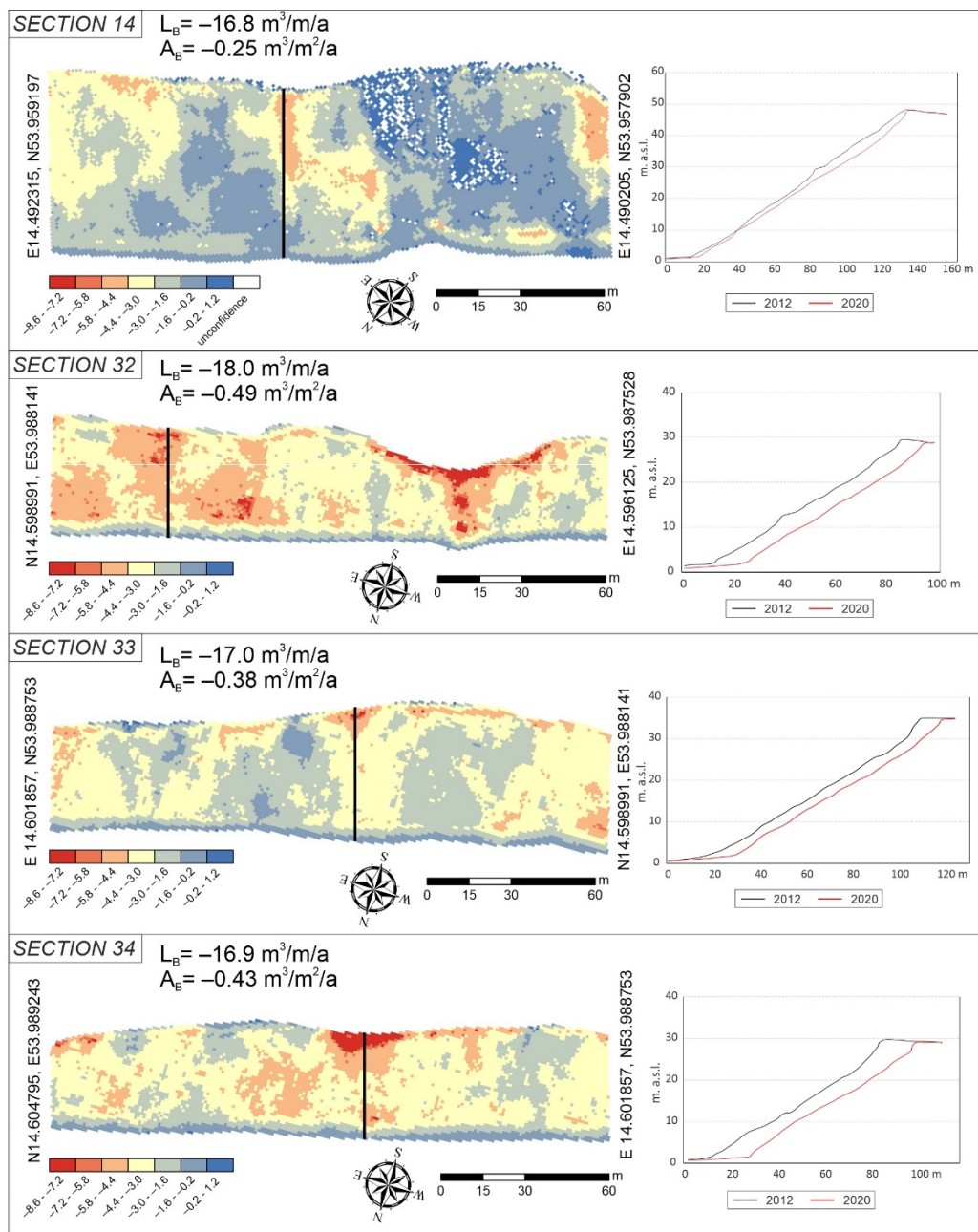

**Figure 11.** Spatial differentiation of the volume changes of the cliff in the sectors subjected to the greatest erosion in the period 2012–2020. Raw data source—Maritime Office in Szczecin. Raw data source—Maritime Office in Szczecin.

**Table 6.** Sediment losses during the observation periods. Raw data source—Maritime Office in Szczecin.

| Period | $T_B$ [m³] | $L_B$ (m³/m/a) | | | $A_B$ (m³/m²/a) | | |
|---|---|---|---|---|---|---|---|
| | | **Mean** | **Min** | **Max** | **Mean** | **Min** | **Max** |
| 2012–2015 | −79,904 | −2.8 | 1.6 | −10.5 | −0.06 | 0.03 | −0.27 |
| 2015–2018 | −268,346 | −9.3 | −0.06 | −30 | −0.22 | 0.00 | −0.66 |
| 2018–2020 | −168,090 | −8.9 | −0.35 | −25.7 | −0.26 | −0.01 | −0.82 |
| 2012–2020 | −509,909 | −6.6 | −1.3 | −18 | −0.17 | −0.03 | −0.49 |

## 4. Discussion

### 4.1. Factors Controling the Erosion of the Cliffs of the Wolin Island

The size of the cliff erosion is primarily determined by the dynamics of the sea (including the frequency of storm surges, the height of the significant wave, and the prevailing direction of the wave approach to the shore). The energy of the waves degrades the cliffs with varying degrees of intensity, which largely depends on the geological structure of the cliff. The conducted research has shown that the greatest erosion occurs on sandy cliffs, and the smallest on clay cliffs. The degree of development of colluvial sediments, constituting a buffer zone of cliff erosion by sea activity, has a very large impact on the volume of changes in the volume of the cliffs. Colluvial sediments are a deposition zone for sediments transported from the upper parts of the cliff. The development of colluvial sediments can take place in two ways and is related to the hydrometeorological conditions. In the first case, during storm episodes, the colluvial sediments are rapidly accumulated at the foot of the cliff as a result of mass movements [20,37,50,75–77]. In the second case, under conditions of weak and moderate sea dynamics, eroded sediments are slowly and systematically deposited at the foot of the cliff along with rainwater runoff and as a result of aeolian processes [78,79]. The accumulated colluvial forms, which take the form of alluvial fans and talus cones at the foot of the cliff, are characterized by a significantly reduced resistance to erosive processes caused by in situ sediments [80,81]. In addition, colluvial forms are developed towards the sea, which is why they are the first to be subject to sea erosion [37]. It should also be noted that high water levels are not required for increased sea erosion of colluviums. This relationship was very well illustrated by the event from the first time period (2012–2015), when within Sector 19 with well-developed colluvial forms, despite the low sea dynamics and clay cliff structure, there was a relatively large loss of sediments. A similar situation occurred in the last period (2018–2020), which was drought especially in the summer seasons. These conditions were conducive to a very good expansion of dump cones on sandy cliffs (especially within Sector 14). Two storm surges were enough for all the material gathered at the foot of the cliff to be carried out to the sea.

Another important factor determining the scale of cliff erosion is the exposure of the shoreline to the dominant direction of wave inflow during strong storm surges [82,83]. The analyzed fragment of the cliff coast of Wolin Island is characterized by a moderately diversified exposure of the coastline. In the western part (Sectors 1–17), the exposure of the coastline ranges from 317° to 320°, then it breaks down in Sectors 18 and 19 and until the very end it ranges from 342° to 344°. The impact of shoreline exposure on shore erosion was particularly visible in the second time period (2015–2018), during which the greatest damage occurred during the storm surge in January 2017 (Alex hurricane). Then the wave for 80% of the storm surge duration (47 h) approached the shore towards the SSW (average 206°), perpendicular to the eastern part of the Wolin Island coast, where the erosion dynamics wetr particularly high.

### 4.2. The Dynamics of the Cliff Coast of the Wolin Island in Comparison to the Shores of the World

The conducted research showed that in the analyzed period 2012–2020, the cliff erosion was equal to $L_B$ = 6.6 m$^3$/m/a $\pm$ 0.3 m$^3$/m/a of sediments, which was $A_B$ = 0.17 m$^3$/m$^2$/a $\pm$ 0.01 m$^3$/m$^2$/a per the area of the analyzed area. These values can be compared to previous studies both on the Wolin Island and in other parts of the South Baltic coast in Poland and on cliffs located in other parts of the world.

The first LiDAR research on the cliffs of the Wolin Island was carried out by [19]. Analyses carried out in one year (2011–2012) on two sites showed that erosion in the case of the area index ($A_B$) was 0.18–0.21 m$^3$/m$^2$/a. Unfortunately, the authors of the study did not provide the length of the analyzed sections, which made it impossible to calculate the ($L_B$) indicator. The results reported by [19] are very similar to the results obtained in this article.

Later studies on the Wolin Island cliff coast showed the average annual erosion in the period 2008–2011 at the level of 5.5 m$^3$/m/a [26]. In this case, the lack of surface data made it impossible to estimate the ($A_B$) indicator.

Subsequent case studies focused on a series of storm surges [84]. The research carried out for a period of 5 months has shown that mainly extreme storm surges are the cause of significant erosion. In the analyzed period, the authors estimated $L_B$ erosion = 50 m$^3$/m. Such a high index largely results from the adopted methodology of the study, in which, in addition to the cliff slope, the beach was also included in the erosion analysis. Therefore, this result is difficult to compare with the results presented in this article. Beach erosion occurs at sea level ≥60 cm N.N., which occurs more frequently and for more hours than cliff erosion level ≥90 cm N.N. [68].

The research on volumetric changes of the Wolin cliffs by various authors was characterized by different temporal and spatial scales, but it should be concluded that the obtained results did not differ significantly from each other and were similar to those presented in this article.

Other studies of the dynamics of moraine cliffs using laser scanning were carried out in the central part of the Polish coast (near Ustka) [4]. Analyses carried out on two sections of cliffs (2 km)—mainly sandy—for a period of 5 years, allowed to determine erosion ranging from $L_B$ = 11–26 m$^3$/m/a. It should be noted that this value is several times higher than that recorded on the cliff coast of the Island of Wolin ($L_B$ = 6.6 ± 0.3 m$^3$/m/a). The cliffs in the vicinity of Ustka are built of sandy sediments that are not very resistant and are easily eroded. Moreover, the central part of the Polish coast is clearly exposed to the open sea with high wave energy. As a result, the region has the highest erosion on the Polish Baltic coast.

Similar studies were carried out on the shores of the Baltic Sea in Germany (Schleswig-Holstein) [23]. Based on the analysis of the 57-km section of the shore built of postglacial sediments (clay and sand), the authors estimated its erosion at the level of $L_B$ = 1.5 m$^3$/m/a. It should be noted that the obtained result is several times lower than that obtained by us. The reason for the lower dynamics of the Schleswig-Holstein cliffs may be the significant isolation of the Belt Sea from the rest of the Baltic Sea, which means that storm surges are less powerful than those occurring on the shores of the South Baltic Sea.

Estimates of the dynamics of cliffs using LiDAR technology in the world literature are well documented. The size of cliff erosion, just like on the shores of the Baltic Sea, is mainly determined by the dynamics of the sea and the geological structure of the coast.

Very similar results of coastal cliff erosion were obtained by [24]. Studies conducted on the south-west coast of England on two 0.3 km-long sites composed of sandstone and post-glacial sediment showed that the ($L_B$) indicator ranged within limits 5.3–11.0 m$^3$/m/a.

Comparable erosion also occurs on the north-west coast of the USA (Strait of Juan de Fuca) [26], which is also composed of postglacial sediments (clay and sands). In this case, the ($L_B$) indicator was on the level 4.1–7.5 m$^3$/m/a.

The problem in question has also been very well recognized on the California coast [21]. The authors, based on the study of an over 800 km long section of the shore made mostly of solid rocks, estimated the annual erosion at the level of $L_B$ = 2.5 m$^3$/m/a. A slightly lower value was estimated for a much shorter section (2.5 km) (Del Mar) also made of solid rocks, which was $L_B$ = 1.3 m$^3$/m/a [12]. The differences in the results could be caused by the different spatial and temporal scales of the research conducted.

Research on the dynamics of cliffs with the use of LiDAR technology was also conducted in other morphoclimatic zones. An example is the study where ice-rich permafrost coasts were analyzed in northern Canada [85]. The calculated index ($L_B$) was at the level of 28 m$^3$/m/a and was several times higher than that obtained in this article. In this case, it should be emphasized that the high dynamics of the cliffs in the polar regions are mainly due to the extensive mass movements that commonly occur during the summer period during the thawing of permafrost.

The quantitative values of the morphodynamics of the cliffs of the Wolin Island, based on remote registration, presented in this article are a detailed description of the knowledge on the rate of erosion of sea coasts. The obtained values do not differ significantly from other studies on the Baltic moraine cliffs and are comparable with other types of cliffs. Unfortunately, due to the use of different methodologies and different budget sheet indicators in various studies, it makes it impossible to conduct a full comparative study. Often, the results presented in the literature have to be converted based on the information provided regarding the length and area of the analyzed cliffs (if such information has been provided).

## 5. Conclusions

The use of ALS laser scanning data enabled the calculation and visualization of the spatial and temporal variability of the morphodynamics of the Wolin Island cliff coast. Accurate altitude data made it possible to quantify the sediment budget, which is valuable information about the state of the coast and its development trends.

The conducted research showed that the volume of cliff changes was determined primarily by the height and frequency of storm surges as well as the geological structure and morphometry of the cliffs. The smallest loss of sediment was observed in the first time period, 2012–2015, during which there were only two events with exceeding the erosive sea level $\geq$90 cm (the maximum sea level was 100 cm). In turn, the greatest loss of sediments occurred in 2015–2018, when five events were recorded with exceeding the erosive sea level $\geq$90 cm (the maximum sea level was 140 cm N.N.). A significant loss of cliff sediments was also found in the period 2018–2020, in which there were two events exceeding the erosive sea level $\geq$90 cm (the maximum sea level was 133 cm N.N.). It can be concluded that, apart from the number of storm surges, the sea level also has a very large impact on the dynamics of the cliffs, especially above the erosion level $\geq$90 cm [68].

In turn, the spatial intensity of erosion processes was most influenced by the geological structure, the degree of development of colluvial forms and the exposure of the coastline. High geomorphological activity was observed mainly on the cliffs, whose structure was dominated by sandy sediments, not very resistant to erosive processes.

The conducted research showed that in the eastern part the index ($A_B$) was almost twice as high ($-0.25$ m$^3$/m$^2$/a) than in the western part ($-0.15$ m$^3$/m$^2$/a), which was mainly due to the greater height of the cliffs in the western part.

A comparative study of the erosion of the Wolin cliffs with other shores of the Baltic Sea showed that the Wolin cliffs are eroded several times faster than the German cliffs (Schlezwig-Holstein) [20] and several times slower than the cliffs of the central Polish coast [3]. Similar sizes of erosion are recorded on other coasts built of postglacial sediments, e.g., England, USA (Juan De Fuca) [23]. On the other hand, the lowest erosion occurs in the case of resistant rock cliffs, e.g., USA (California) [18,19].

Many studies based on LiDAR technology have been developed in the study of the dynamics of cliffs. Often, however, the presented results—due to the individuality of the research problem—are difficult to compare with each other. The results are often presented in the form of a total budget ($T_B$) for an unspecified length or coastal area, or refer only to specific areas of a cliff (e.g., landslides) or a beach (counted in the budget of a cliff coast). Therefore, to facilitate comparative studies of cliffs in different regions, the authors of this article proposed the adoption of three basic indicators regarding the degree of coastal activity: one general—total budget ($T_B$); and two normalized—length-normalized budget ($L_B$) and area-normalized budget ($A_B$).

The presented example of the use of data (ALS) to determine the temporal and spatial morphodynamics of the cliff coast indicates a very high usefulness of remote sensing data, especially for environmental analyses on a large spatial scale. The obtained results have not only great scientific value, but also application value, related to—among others—maritime coastal zone management.

**Supplementary Materials:** The following supporting information can be downloaded at: https://www.mdpi.com/article/10.3390/rs14133115/s1, Table S1: Parameters of individual sectors.

**Author Contributions:** Conceptualization, M.W. and J.T.; Methodology, M.W. and J.T.; Software, M.W.; Data curation, M.W.; Writing—original draft preparation, M.W., J.T. and M.H.; Writing—review and editing, M.W. and M.H.; Visualization, M.W. All authors have read and agreed to the published version of the manuscript.

**Funding:** The APC was funded by Adam Mickiewicz University in Poznan.

**Data Availability Statement:** The data presented in this study are available on the request from the corresponding author.

**Acknowledgments:** Hydrometeorological data were obtained from the Institute of Meteorology and Water Management, National Research Institute in Warsaw. LiDAR data were obtained from Maritime Office in Szczecin.

**Conflicts of Interest:** The authors declare no conflict of interest.

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
