# Peer review of "Assessment of Moraine Cliff Spatio-Temporal Erosion on Wolin Island Using ALS Data Analysis"

_remotesensing, doi:10.3390/rs14133115_

Round 1

Reviewer 1 Report

Review of the manuscript ID remotesensing-1746589 entitled ‘Temporal and spatial variability of morphodynamics of the moraine cliff of the Wolin Island in the light of ALS measurements’

Based on four series of airborne laser scanning data (2012;2015;2018;2020) obtained from the Maritime Office in Szczecin, the authors performed a differential comparative analysis of changes in the topographic surface of an eroding cliff coast on Wolin Island. Classified ALS data formed the basis for generating four DTM models, most likely with 1m cells. Using GMC software, the authors determined surface changes, which allowed estimating the erosion-deposition balance. Changes in surface area were related to the type of surface sediment, which made it possible to determine that in the analysed period 2012-2020, the greatest erosion occurred on sandy cliffs, while the least erosion occurred on clay cliffs and on cliffs densely covered with vegetation. The authors proposed the introduction of two indicators of coastal erosion in sediment budget studies: (1) length-normalized sediment budget (LB) and expressed in the unit m3/m and (2) area-normalized sediment budget (AB) expressed in the unit m3/m2.

The proposed topic is interesting and fits well with the theme of the proposed remote sensing special issue. However, the text contains a number of shortcomings that should be reviewed and corrected before considering acceptance of this manuscript for publication. First, the title. In my opinion, it is overloaded. Morphodynamics (if the authors evaluate morphodynamics at all) versus variability is too much. One of these terms (either variability of morphology vs. gemorphic changes or morphodynamics) would suffice. The title is somewhat disjointed and too long. Also, the last part is questionable. The authors do not analyse ALS measurements, but only use the ALS data. I suggest to simplify and clarify the topic e.g. Assessment of moraine cliff spatio-temporal morphodynamics on Wolin Island using ALS data analysis. As long as indeed the topic of this paper is the estimation of cliff coastal morphodynamics.

The first major problem is the imprecise methodological chapter. First, the authors do not provide sufficient characterization of the ALS data. There is no information about positioning and registration error. Further in the text of the manuscript, an error magnitude of 0.15 (no unit) appears. In general, as long as it is 0.15 m, it is a plausible magnitude, but the data should be different for each raid and specified by the data provider. This should be supplemented. In addition, when characterizing ALS data, authors provide the distance between points. Usually it is the points density [pts/m2]. The process of DTMs generation is not described at all. This is crucial for the whole DoD analysis. Neither the method nor the software used to perform this process is given. We do not know what coordinate system the data were in. We do not know what size the cell was or why it was that size. We do not know how many points were used to average cell height. In the further part of the text Authors mention 1m. If this is the size of the cell then on a steep cliff the height of the cell can be averaged from points with height difference of several meters. This process should be described extremely carefully. Further, not much has been written about GCD analysis. In the description of uncertainty estimation, the authors describe the process of estimating the RMSE for differential DTMs. This part is incomprehensible to me. While RMSE for individual cells would have some justification (in the context of large expected differences between cloud points), for whole models I see no sense or justification. The GCD software provides the ability to account for error and estimate uncertainty at two stages: 1) as an error Surface and 2) during DoD generation. In the second step there are three options: 1) entering a simple minimum level of detection (threshold), which I believe should be 0.15 (or the default 0.2 in the context of the previous comments on ALS data errors) 2) propagated errors and 3) probabilistic thresholding. The choice of the best method is up to the authors; nevertheless, both the choice and its justification should be described in the methodology. Generally, the method section (or in fact it should be materials and methods) should be rewritten (completed and organized. It would be advisable to show all stages of the analysis on a flowchart. Another aspect is the indicators proposed by the authors:  1) sediment budget normalized by length (LB) and expressed in unit m3/m and 2) sediment budget normalized by area (AB) expressed in unit m3/m2. They do not appeal to me. The motives for their introduction are not clearly spelled out. What do they bring new? The second (AB) is a commonly used rate indicator expressed in m (m3/m2=m). The rate is commonly used for erosion and deposition dynamics and ensures comparability of results. What does modifying it and presenting it as m3/m2 provide? As for the first one, it convinces me even less. Is there any relationship between coastal length and sediment volume that would justify such interpretations? If the authors insist on using these indices, they should justify their usefulness better. Closing the topic of methodological shortcomings , I would like to refer to the DoD analysis area No full map of the area with DoD results in the paper. The description of the methodology does not specify whether the Authors used a mask to dance off the Area of Interest (AoI). If not then how did they ensure comparability of DTMs. In my opinion, a figure showing the analysis area and AoI mask is necessary. The paper is not properly illustrated. There are no figures showing surface differences of all analysed coastal fragments. Only sections were depicted without showing where they are located.

In addition, there are many language errors in the text. Even in two adjacent sentences ALS is once developed as airborne and once as aerial laser scanning. DEM is described as a digital height model. Sediment losses in relation to volume changes estimated from elevation differences. Caption: Research area with visible cliff-building sediments, is quite peculiar. In general, the text should be proofread preferably by an English native speaker familiar with geomorphological terminology. In summary, although the proposed topic is interesting and well suited to the IS, it still requires considerable work both in terms of improving the substantive content as well as editorial and linguistic correctness. I have applied the detailed comments directly to the pdf.

Author Response

First, we would like to thank you for a very valuable and detailed review that allowed us to significantly improve the methodology of our article. We tried to correct the indicated errors and clarify the ambiguities. Below are the answers to your review.

  1. Overloaded title

Thank you for paying attention to this issue. Indeed, the title is too overloaded and does not fully reflect the main goal of the presented research. Taking into account your proposal, we decided to modify the title as follows: Assessment of moraine cliff spatio-temporal erosion on Wolin Island using ALS data analysis. We have replaced the expression morphodynamics with erosion because this better reflects the main goal of our study.

  1. No information about positioning and registration error

Unfortunately, we do not have such an information because the data provider did not provide it to us. We have repeatedly asked for this information, but received no reply. The only information we received was that the position accuracy is not worse than 0.15 m. This general information prompted us to look for the accuracy directly on DEM models based on the differences in the height of stable reference points in DEM pairs (2012-2015, 20215-2018, 2018-2020).

  1. Point density

This parameter was recalculated and supplemented in the table at the point spacing site. It applies only to the points of the ground class, because on their values the DEM models were built.

  1. DEM generating process

The models were built on the basis of the value of class 2 (ground) points. DEM models were made in the ArcGis program (conversion tool: Las dataset to raster) in the coordinate system (ETRS_1989_Poland_CS92). The point cloud density analysis (Table 1) was the basis for determining the raster resolution. It is assumed that at least 2 points are required to determine the cell value, assuming that more of them increasing the accuracy of the cell value. Finally, it was decided to build a raster with a resolution of 1m. The binning interpolation method was used to determine the cell value, taking into account the mean value of the points lying within it, which was the optimal solution in the case of the analysis of steep slopes. Using a minimum value on steep slopes could underestimate the actual height.

  1. The description of the GCD analysis

In the first stage of the analyzes, it was necessary to import DEM models to the Geomorphic Change Detection (GCD) program. Then, the models were subjected to differential analysis, indicating the simple minimum level of detection based on an earlier analysis of reference points. The calculated volume differences were limited to the range of the previously prepared AOI (Area of Interst) masks (range from the cliff top line to the cliff foot line along the entire length of the analyzed coast) (Figure 3). Due to the fact that the surface of the slope was different in the subsequent observation periods, AOI masks were prepared for each of the performed differential analyzes (4 in total). Then each mask was divided into 200 m long sectors and used for budget segregation in order to capture the spatial variability of volume changes.

As a result, the total net volume difference was calculated in m3 and its two components: erosion (total volume of surface lowering) and deposition treated as an accumulation of sediments (total volume of surface rising). Additionally, when indicating the minimum level of detection, the volume error and its percentage share are calculated for each of the calculated indicators. The results are visualized in the form of differential models indicating the spatial differentiation of erosion and deposition zones. Surfaces with values in the uncertainty range are assigned the NoData value.

  1. Budget indicators

The presented indicators are a proposal for the standardization of the results obtained in volumetric analyzes of coastal erosion. Thanks to such standardization, it will be possible to compare the results obtained in different regions of the world. Currently, the literature often gives the results in the form of a total balance without the reference to the spatial unit in which the analyzes were conducted. This information is valuable but of little use for comparative analyzes. The description of the indicators has been made more specific in the text to some extent.

  1. RMSE error for pairs of DEM models

The knowledge of the accuracy of the raster cell values was very important for the volumetric analysis. This information was important to determine the minimum level of detection in later volumetric analyzes. For this purpose, stable objects (whose position and height did not change over time) were selected as reference points on raster models. The elevation root mean square error of the reference points determined the minimum level of detection. In total, 28 reference points were selected for the analysis (15 points on the western section and 13 points on the eastern section) (Figure 3). Elevation RMSE analysis were determined separately for the eastern and western parts in pairs for the years covered by the analysis (2012-2015, 2015-2018, 2018-2020, 2012-2020). The obtained values were within the limits of W: 0.05 m - 0.1 m and E: 0.07 m - 0.1 m

  1. No full map of the area with DoD results

We intentionally resigned from presenting the full map due to its illegibility (the figure presenting 9 km of coast with an average height of 30-40m is completely unreadable). We think that a better solution of this problem will be to visualize indicators in individual sectors. For this purpose, we have included two figures in the text (fig. 8, 9)

  1. No AoI mask

The calculated volume differences were limited to the range of the previously prepared AOI (Area of Interst) masks (range from the cliff top line to the cliff foot line along the entire length of the analyzed coast) (Figure 3). Due to the fact that the surface of the slope was different in the subsequent observation periods, AOI masks were prepared for each of the performed differential analyzes (4 in total). Then each mask was divided into 200 m long sectors and used for budget segregation in order to capture the spatial variability of volume changes.

  1. Ambiguities and language errors

All indicated ambiguities and linguistic errors have been removed

  1. References

All missing references have been filled

  1. In addition

In the methodological chapter, the order of the sections was changed. Due to the fact that in the part related to data analyzes, the AoI map had to be presented on the background of the map of the research area, we decided that the research area should be presented first. Therefore, the research area section has been moved to the beginning of the methodological chapter 2.1

Reviewer 2 Report

Dear authors, you are presenting a neat and very nice paper.

I only have a few very minor corrections:

1. Figure 1. Bottom panel, add geographic coordinates

2. Figure 7. Could you add geographic coordinates?

Moreover, I like the paper, is very well presented and explained.

All the best.

Author Response

Thank you very much for your tips. All the comments concerning the figures have been taken into account

Best regards

Marcin Winowski

Reviewer 3 Report

Dear authors,

Your work is interesting, and the data included in this ms are significant. However, I have remarked that the technical specification of the used ALS is missing. It is important for the readers to understand the accuracy of your data. Also, the ground control points are missing. Before you send back your paper please discuss which is the main message of your work and what is the main result addressed. Please also see my annotated version it includes two additional references and some other comments.

Author Response

First, we would like to thank you for your review. Below are the answers to your review

  1. Point cloud specification

Unfortunately, we do not have information about accuracy of point clouds because the data provider did not provide it to us. We have repeatedly asked for this information, but received no reply. The only information we received was that the position accuracy is not worse than 0.15 m. This general information prompted us to look for the accuracy directly on DEM models based on the differences in the height of stable reference points in DEM pairs (2012-2015, 20215-2018, 2018-2020).

We only suplemented point cloud density for the ground class, because on their values the DEM models were built

  1. Ground control points

Information on GCP has been supplemented in the text (number of points), their spatial distribution on the map has also been indicated

  1. Main massage and result of work

We have completed this information at the end of the introduction section

  1. References

All missing references have been filled

  1. Figures captions

All indicated captions were corrected

Best regards

Marcin Winowski

Jacek Tylkowski

Marcin Hojan

Round 2

Reviewer 1 Report

The manuscript has been revised and missing information completed. The Authors have also added new figures. However, they did not avoid minor errors. In the newly added figures (Figs. 1, 3, 8, 9) the decimal separator should be changed from a comma to a dot in the description of linear and elevation scales (a comma is a thousandths separator). However, this is a minor correction that can be done during proofreading check.